# Pre-training of Recurrent Neural Networks via Linear Autoencoders

**Luca Pasa, Alessandro Sperduti**
Department of Mathematics
University of Padova, Italy
{pasa,sperduti}@math.unipd.it

## Abstract

We propose a pre-training technique for recurrent neural networks based on linear autoencoder networks for sequences, i.e. linear dynamical systems modelling the target sequences. We start by giving a closed form solution for the definition of the optimal weights of a linear autoencoder given a training set of sequences. This solution, however, is computationally very demanding, so we suggest a procedure to get an approximate solution for a given number of hidden units. The weights obtained for the linear autoencoder are then used as initial weights for the input-to-hidden connections of a recurrent neural network, which is then trained on the desired task. Using four well known datasets of sequences of polyphonic music, we show that the proposed pre-training approach is highly effective, since it allows to largely improve the state of the art results on all the considered datasets.

## 1 Introduction

Recurrent Neural Networks (RNN) constitute a powerful computational tool for sequences modelling and prediction [1]. However, training a RNN is not an easy task, mainly because of the well known vanishing gradient problem which makes difficult to learn long-term dependencies [2]. Although alternative architectures, e.g. LSTM networks [3], and more efficient training procedures, such as Hessian Free Optimization [4], have been proposed to circumvent this problem, reliable and effective training of RNNs is still an open problem.

The vanishing gradient problem is also an obstacle to Deep Learning, e.g., [5, 6, 7]. In that context, there is a growing evidence that effective learning should be based on relevant and robust internal representations developed in autonomy by the learning system. This is usually achieved in vectorial spaces by exploiting nonlinear autoencoder networks to learn rich internal representations of input data which are then used as input to shallow neural classifiers or predictors (see, for example, [8]). The importance to start gradient-based learning from a good initial point in the parameter space has also been pointed out in [9]. Relationship between autoencoder networks and Principal Component Analysis (PCA) [10] is well known since late '80s, especially in the case of linear hidden units [11, 12]. More recently, linear autoencoder networks for structured data have been studied in [13, 14, 15], where an exact closed-form solution for the weights is given in the case of a number of hidden units equal to the rank of the full data matrix.

In this paper, we borrow the conceptual framework presented in [13, 16] to devise an effective pre-training approach, based on linear autoencoder networks for sequences, to get a good starting point into the weight space of a RNN, which can then be successfully trained even in presence of long-term dependencies. Specifically, we revise the theoretical approach presented in [13] by: *i)* giving a simpler and direct solution to the problem of devising an exact closed-form solution (full rank case) for the weights of a linear autoencoder network for sequences, highlighting the relationship between the proposed solution and PCA of the input data; *ii)* introducing a new formulation of

the autoencoder learning problem able to return an optimal solution also in the case of a number of hidden units which is less than the rank of the full data matrix; *iii)* proposing a procedure for approximate learning of the autoencoder network weights under the scenario of very large sequence datasets. More importantly, we show how to use the linear autoencoder network solution to derive a good initial point into a RNN weight space, and how the proposed approach is able to return quite impressive results when applied to prediction tasks involving long sequences of polyphonic music.

## 2   Linear Autoencoder Networks for Sequences

In [11, 12] it is shown that principal directions of a set of vectors $\mathbf{x}_i \in \mathbb{R}^k$ are related to solutions obtained by training linear autoencoder networks

$$\mathbf{o}_i = \mathbf{W}_{output}\mathbf{W}_{hidden}\mathbf{x}_i, \ i = 1, \dots, n, \tag{1}$$

where $\mathbf{W}_{hidden} \in \mathbb{R}^{p \times k}$, $\mathbf{W}_{output} \in \mathbb{R}^{k \times p}$, $p \ll k$, and the network is trained so to get $\mathbf{o}_i = \mathbf{x}_i$, $\forall i$.

When considering a temporal sequence $\mathbf{x}_1, \mathbf{x}_2, \dots, \mathbf{x}_t, \dots$ of input vectors, where $t$ is a discrete time index, a linear autoencoder can be defined by considering the coupled linear dynamical systems

$$\mathbf{y}_t = \mathbf{A}\mathbf{x}_t + \mathbf{B}\mathbf{y}_{t-1} \quad (2) \qquad\qquad \begin{bmatrix} \mathbf{x}_t \\ \mathbf{y}_{t-1} \end{bmatrix} = \mathbf{C}\mathbf{y}_t \quad (3)$$

It should be noticed that eqs. (2) and (3) extend the linear transformation defined in eq. (1) by introducing a *memory term* involving matrix $\mathbf{B} \in \mathbb{R}^{p \times p}$. In fact, $\mathbf{y}_{t-1}$ is inserted in the right part of equation (2) to keep track of the input history through time: this is done exploiting a state space representation. Eq. (3) represents the decoding part of the autoencoder: when a state $\mathbf{y}_t$ is multiplied by $\mathbf{C}$, the observed input $\mathbf{x}_t$ at time $t$ and state at time $t-1$, i.e. $\mathbf{y}_{t-1}$, are generated. Decoding can then continue from $\mathbf{y}_{t-1}$. This formulation has been proposed, for example, in [17] where an iterative procedure to learn weight matrices $\mathbf{A}$ and $\mathbf{B}$, based on Oja's rule, is presented. No proof of convergence for the proposed procedure is however given. More recently, an exact closed-form solution for the weights has been given in the case of a number of hidden units equal to the rank of the full data matrix (full rank case) [13, 16]. In this section, we revise this result. In addition, we give an exact solution also for the case in which the number of hidden units is strictly less than the rank of the full data matrix.

The basic idea of [13, 16] is to look for directions of high variance into the *state space* of the dynamical linear system (2). Let start by considering a single sequence $\mathbf{x}_1, \mathbf{x}_2, \dots, \mathbf{x}_t, \dots, \mathbf{x}_n$ and the state vectors of the corresponding induced state sequence collected as rows of a matrix $\mathbf{Y} = [\mathbf{y}_1, \mathbf{y}_2, \mathbf{y}_3, \cdots, \mathbf{y}_n]^\mathsf{T}$. By using the initial condition $\mathbf{y}_0 = \mathbf{0}$ (the null vector), and the dynamical linear system (2), we can rewrite the $\mathbf{Y}$ matrix as

$$\mathbf{Y} = \underbrace{\begin{bmatrix} \mathbf{x}_1^\mathsf{T} & \mathbf{0} & \mathbf{0} & \mathbf{0} & \cdots & \mathbf{0} \\ \mathbf{x}_2^\mathsf{T} & \mathbf{x}_1^\mathsf{T} & \mathbf{0} & \mathbf{0} & \cdots & \mathbf{0} \\ \mathbf{x}_3^\mathsf{T} & \mathbf{x}_2^\mathsf{T} & \mathbf{x}_1^\mathsf{T} & \mathbf{0} & \cdots & \mathbf{0} \\ \vdots & \vdots & \vdots & \vdots & \vdots & \vdots \\ \mathbf{x}_n^\mathsf{T} & \mathbf{x}_{n-1}^\mathsf{T} & \mathbf{x}_{n-2}^\mathsf{T} & \cdots & \mathbf{x}_2^\mathsf{T} & \mathbf{x}_1^\mathsf{T} \end{bmatrix}}_{\mathbf{\Xi}} \underbrace{\begin{bmatrix} \mathbf{A}^\mathsf{T} \\ \mathbf{A}^\mathsf{T}\mathbf{B}^\mathsf{T} \\ \mathbf{A}^\mathsf{T}\mathbf{B}^{2\mathsf{T}} \\ \vdots \\ \mathbf{A}^\mathsf{T}\mathbf{B}^{n-1\mathsf{T}} \end{bmatrix}}_{\mathbf{\Omega}}$$

where, given $s = kn$, $\mathbf{\Xi} \in \mathbb{R}^{n \times s}$ is a data matrix collecting all the (inverted) input subsequences (including the whole sequence) as rows, and $\mathbf{\Omega}$ is the parameter matrix of the dynamical system.

Now, we are interested in using a state space of dimension $p \ll n$, i.e. $\mathbf{y}_t \in \mathbb{R}^p$, such that as much information as contained in $\mathbf{\Omega}$ is preserved. We start by factorizing $\mathbf{\Xi}$ using SVD, obtaining $\mathbf{\Xi} = \mathbf{V}\mathbf{\Lambda}\mathbf{U}^\mathsf{T}$ where $\mathbf{V} \in \mathbb{R}^{n \times n}$ is an unitary matrix, $\mathbf{\Lambda} \in \mathbb{R}^{n \times s}$ is a rectangular diagonal matrix with nonnegative real numbers on the diagonal with $\lambda_{1,1} \geq \lambda_{2,2} \geq \cdots \geq \lambda_{n,n}$ (the singular values), and $\mathbf{U}^\mathsf{T} \in \mathbb{R}^{s \times n}$ is a unitary matrix.

It is important to notice that columns of $\mathbf{U}^\mathsf{T}$ which correspond to nonzero singular values, apart some mathematical technicalities, basically correspond to the principal directions of data, i.e. PCA.

If the rank of $\mathbf{\Xi}$ is $p$, then only the first $p$ elements of the diagonal of $\mathbf{\Lambda}$ are not null, and the above decomposition can be reduced to $\mathbf{\Xi} = \mathbf{V}^{(p)}\mathbf{\Lambda}^{(p)}\mathbf{U}^{(p)\mathsf{T}}$ where $\mathbf{V}^{(p)} \in \mathbb{R}^{n \times p}$, $\mathbf{\Lambda}^{(p)} \in \mathbb{R}^{p \times p}$,

and $\mathbf{U}^{(p)}{}^{\mathsf{T}} \in \mathbb{R}^{p \times n}$. Now we can observe that $\mathbf{U}^{(p)}{}^{\mathsf{T}} \mathbf{U}^{(p)} = \mathbf{I}$ (where $\mathbf{I}$ is the identity matrix of dimension $p$), since by definition the columns of $\mathbf{U}^{(p)}$ are orthogonal, and by imposing $\mathbf{\Omega} = \mathbf{U}^{(p)}$, we can derive "optimal" matrices $\mathbf{A} \in \mathbb{R}^{p \times k}$ and $\mathbf{B} \in \mathbb{R}^{p \times p}$ for our dynamical system, which will have corresponding state space matrix $\mathbf{Y}^{(p)} = \mathbf{\Xi}\mathbf{\Omega} = \mathbf{\Xi}\mathbf{U}^{(p)} = \mathbf{V}^{(p)}\mathbf{\Lambda}^{(p)}\mathbf{U}^{(p)}{}^{\mathsf{T}}\mathbf{U}^{(p)} = \mathbf{V}^{(p)}\mathbf{\Lambda}^{(p)}$. Thus, if we represent $\mathbf{U}^{(p)}$ as composed of $n$ submatrices $\mathbf{U}_i^{(p)}$, each of size $k \times p$, the problem reduces to find matrices $\mathbf{A}$ and $\mathbf{B}$ such that

$$\mathbf{\Omega} = \begin{bmatrix} \mathbf{A}^{\mathsf{T}} \\ \mathbf{A}^{\mathsf{T}}\mathbf{B}^{\mathsf{T}} \\ \mathbf{A}^{\mathsf{T}}\mathbf{B}^{2\mathsf{T}} \\ \vdots \\ \mathbf{A}^{\mathsf{T}}\mathbf{B}^{n-1\mathsf{T}} \end{bmatrix} = \begin{bmatrix} \mathbf{U}_1^{(p)} \\ \mathbf{U}_2^{(p)} \\ \mathbf{U}_3^{(p)} \\ \vdots \\ \mathbf{U}_n^{(p)} \end{bmatrix} = \mathbf{U}^{(p)}. \tag{4}$$

The reason to impose $\mathbf{\Omega} = \mathbf{U}^{(p)}$ is to get a state space where the coordinates are uncorrelated so to diagonalise the empirical sample covariance matrix of the states. Please, note that in this way each state (i.e., row of the $\mathbf{Y}$ matrix) corresponds to a row of the data matrix $\mathbf{\Xi}$, i.e. the unrolled (sub)sequence read up to a given time t. If the rows of $\mathbf{\Xi}$ were vectors, this would correspond to compute PCA, keeping only the fist $p$ principal directions.

In the following, we demonstrate that there exists a solution to the above equation. We start by observing that $\mathbf{\Xi}$ owns a special structure, i.e. given $\mathbf{\Xi} = [\mathbf{\Xi}_1 \ \mathbf{\Xi}_2 \ \cdots \ \mathbf{\Xi}_n]$, where $\mathbf{\Xi}_i \in \mathbb{R}^{n \times k}$, then for $i = 1, \ldots, n-1$, $\mathbf{\Xi}_{i+1} = \mathbf{R}_n\mathbf{\Xi}_i = \begin{bmatrix} \mathbf{0}_{1 \times (n-1)} & \mathbf{0}_{1 \times 1} \\ \mathbf{I}_{(n-1) \times (n-1)} & \mathbf{0}_{(n-1) \times 1} \end{bmatrix} \mathbf{\Xi}_i$, and $\mathbf{R}_n\mathbf{\Xi}_n = \mathbf{0}$, i.e. the null matrix of size $n \times k$. Moreover, by singular value decomposition, we have $\mathbf{\Xi}_i = \mathbf{V}^{(p)}\mathbf{\Lambda}^{(p)}\mathbf{U}_i^{(p)}{}^{\mathsf{T}}$, for $i = 1, \ldots, n$. Using the fact that $\mathbf{V}^{(p)}{}^{\mathsf{T}}\mathbf{V}^{(p)} = \mathbf{I}$, and combining the above equations, we get $\mathbf{U}_{i+t}^{(p)} = \mathbf{U}_i^{(p)}\mathbf{Q}^t$, for $i = 1, \ldots, n-1$, and $t = 1, \ldots, n-i$, where $\mathbf{Q} = \mathbf{\Lambda}^{(p)}\mathbf{V}^{(p)}{}^{\mathsf{T}}\mathbf{R}_n^{\mathsf{T}}\mathbf{V}^{(p)}\mathbf{\Lambda}^{(p)}{}^{-1}$. Moreover, we have that $\mathbf{U}_n^{(p)}\mathbf{Q} = \mathbf{0}$ since $\mathbf{U}_n^{(p)}\mathbf{Q} = \mathbf{U}_n^{(p)}\mathbf{\Lambda}^{(p)}\mathbf{V}^{(p)}{}^{\mathsf{T}}\mathbf{R}_n^{\mathsf{T}}\mathbf{V}^{(p)}\mathbf{\Lambda}^{(p)}{}^{-1} = \underbrace{(\mathbf{R}_n\mathbf{\Xi}_n)}_{=\mathbf{0}}{}^{\mathsf{T}}\mathbf{V}^{(p)}\mathbf{\Lambda}^{(p)}{}^{-1}$. Thus, eq. (4) is satisfied by

$\mathbf{A} = \mathbf{U}_1^{(p)}{}^{\mathsf{T}}$ and $\mathbf{B} = \mathbf{Q}^{\mathsf{T}}$. It is interesting to note that the original data $\mathbf{\Xi}$ can be recovered by computing $\mathbf{Y}^{(p)}\mathbf{U}^{(p)}{}^{\mathsf{T}} = \mathbf{V}^{(p)}\mathbf{\Lambda}^{(p)}\mathbf{U}^{(p)}{}^{\mathsf{T}} = \mathbf{\Xi}$, which can be achieved by running the system

$$\begin{bmatrix} \mathbf{x}_t \\ \mathbf{y}_{t-1} \end{bmatrix} = \begin{bmatrix} \mathbf{A}^{\mathsf{T}} \\ \mathbf{B}^{\mathsf{T}} \end{bmatrix} \mathbf{y}_t$$

starting from $\mathbf{y}_n$, i.e. $\begin{bmatrix} \mathbf{A}^{\mathsf{T}} \\ \mathbf{B}^{\mathsf{T}} \end{bmatrix}$ is the matrix $\mathbf{C}$ defined in eq. (3).

Finally, it is important to remark that the above construction works not only for a single sequence, but also for a set of sequences of different length. For example, let consider the two sequences $(\mathbf{x}_1^a, \mathbf{x}_2^a, \mathbf{x}_3^a)$ and $(\mathbf{x}_1^b, \mathbf{x}_2^b)$. Then, we have

$$\mathbf{\Xi}_{\mathbf{a}} = \begin{bmatrix} \mathbf{x}_1^{a\mathsf{T}} & \mathbf{0} & \mathbf{0} \\ \mathbf{x}_2^{a\mathsf{T}} & \mathbf{x}_1^{a\mathsf{T}} & \mathbf{0} \\ \mathbf{x}_3^{a\mathsf{T}} & \mathbf{x}_2^{a\mathsf{T}} & \mathbf{x}_1^{a\mathsf{T}} \end{bmatrix} \quad \text{and} \quad \mathbf{\Xi}_{\mathbf{b}} = \begin{bmatrix} \mathbf{x}_1^{b\mathsf{T}} & \mathbf{0} \\ \mathbf{x}_2^{b\mathsf{T}} & \mathbf{x}_1^{b\mathsf{T}} \end{bmatrix}$$

which can be collected together to obtain $\mathbf{\Xi} = \begin{bmatrix} \mathbf{\Xi}_{\mathbf{a}} \\ \hline \mathbf{\Xi}_{\mathbf{b}} & \mathbf{0}_{2 \times 1} \end{bmatrix}$, and $\mathbf{R} = \begin{bmatrix} \mathbf{R}_4 \\ \hline \mathbf{R}_2 & \mathbf{0}_{2 \times 1} \end{bmatrix}$.

As a final remark, it should be stressed that the above construction *only* works if $p$ is equal to the rank of $\mathbf{\Xi}$. In the next section, we treat the case in which $p < rank(\mathbf{\Xi})$.

## 2.1 Optimal solution for low dimensional autoencoders

When $p < rank(\mathbf{\Xi})$ the solution given above breaks down because $\tilde{\mathbf{\Xi}}_i = \mathbf{V}^{(p)}\mathbf{L}^{(p)}\mathbf{U}_i^{(p)}{}^{\mathsf{T}} \neq \mathbf{\Xi}_i$, and consequently $\hat{\mathbf{\Xi}}_{i+1} \neq \mathbf{R}_n\tilde{\mathbf{\Xi}}_i$. So the question is whether the proposed solutions for $\mathbf{A}$ and $\mathbf{B}$ still hold the best reconstruction error when $p < rank(\mathbf{\Xi})$.

In this paper, we answer in negative terms to this question by resorting to a new formulation of our problem where we introduce *slack-like* matrices $\mathbf{E}_i^{(p)} \in \mathbb{R}^{k \times p}$, $i = 1, \ldots, n+1$ collecting the reconstruction errors, which need to be minimised:

$$\min_{\mathbf{Q} \in \mathbb{R}^{p \times p}, \mathbf{E}_i^{(p)}} \sum_{i=1}^{n+1} \|\mathbf{E}_i^{(p)}\|_F^2$$

$$subject \ to: \quad \begin{bmatrix} \mathbf{U}_1^{(p)} + \mathbf{E}_1^{(p)} \\ \mathbf{U}_2^{(p)} + \mathbf{E}_2^{(p)} \\ \mathbf{U}_3^{(p)} + \mathbf{E}_3^{(p)} \\ \vdots \\ \mathbf{U}_n^{(p)} + \mathbf{E}_n^{(p)} \end{bmatrix} \mathbf{Q} = \begin{bmatrix} \mathbf{U}_2^{(p)} + \mathbf{E}_2^{(p)} \\ \mathbf{U}_3^{(p)} + \mathbf{E}_3^{(p)} \\ \vdots \\ \mathbf{U}_n^{(p)} + \mathbf{E}_n^{(p)} \\ \mathbf{E}_{n+1}^{(p)} \end{bmatrix} \tag{5}$$

Notice that the problem above is convex both in the objective function and in the constraints; thus it only has global optimal solutions $\mathbf{E}_i^*$ and $\mathbf{Q}^*$, from which we can derive $\mathbf{A}^\mathsf{T} = \mathbf{U}_1^{(p)} + \mathbf{E}_1^*$ and $\mathbf{B}^\mathsf{T} = \mathbf{Q}^*$. Specifically, when $p = rank(\mathbf{\Xi})$, $\mathbf{R}_{s,k}^\mathsf{T} \mathbf{U}^{(p)}$ is in the span of $\mathbf{U}^{(p)}$ and the optimal solution is given by $\mathbf{E}_i^* = \mathbf{0}_{k \times p} \ \forall i$, and $\mathbf{Q}^* = \mathbf{U}^{(p)\mathsf{T}} \mathbf{R}_{s,k}^\mathsf{T} \mathbf{U}^{(p)}$, i.e. the solution we have already described. If $p < rank(\mathbf{\Xi})$, the optimal solution cannot have $\forall i$, $\mathbf{E}_i^* = \mathbf{0}_{k \times p}$. However, it is not difficult to devise an iterative procedure to reach the minimum. Since in the experimental section we do not exploit the solution to this problem for reasons that we will explain later, here we just sketch such procedure. It helps to observe that, given a fixed $\mathbf{Q}$, the optimal solution for $\mathbf{E}_i^{(p)}$ is given by

$$[\tilde{\mathbf{E}}_1^{(p)}, \tilde{\mathbf{E}}_2^{(p)}, \ldots, \tilde{\mathbf{E}}_{n+1}^{(p)}] = [\mathbf{U}_1^{(p)} \mathbf{Q} - \mathbf{U}_2^{(p)}, \mathbf{U}_1^{(p)} \mathbf{Q}^2 - \mathbf{U}_3^{(p)}, \mathbf{U}_1^{(p)} \mathbf{Q}^3 - \mathbf{U}_4^{(p)}, \ldots] \mathbf{M}_{\mathbf{Q}}^+$$

where $\mathbf{M}_{\mathbf{Q}}^+$ is the pseudo inverse of $\mathbf{M}_{\mathbf{Q}} = \begin{bmatrix} -\mathbf{Q} & -\mathbf{Q}^2 & -\mathbf{Q}^3 & \cdots \\ \mathbf{I} & \mathbf{0} & \mathbf{0} & \cdots \\ \mathbf{0} & \mathbf{I} & \mathbf{0} & \cdots \\ \mathbf{0} & \mathbf{0} & \mathbf{I} & \cdots \\ \vdots & \vdots & \vdots & \vdots \end{bmatrix}$.

In general, $\tilde{\mathbf{E}}^{(p)} = \left[\tilde{\mathbf{E}}_1^{(p)\mathsf{T}}, \tilde{\mathbf{E}}_2^{(p)\mathsf{T}}, \tilde{\mathbf{E}}_3^{(p)\mathsf{T}}, \cdots, \tilde{\mathbf{E}}_n^{(p)\mathsf{T}}\right]^\mathsf{T}$ can be decomposed into a component in the span of $\mathbf{U}^{(p)}$ and a component $\mathbf{E}^{(p)\perp}$ orthogonal to it. Notice that $\mathbf{E}^{(p)\perp}$ cannot be reduced, while (part of) the other component can be absorbed into $\mathbf{Q}$ by defining $\tilde{\mathbf{U}}^{(p)} = \mathbf{U}^{(p)} + \mathbf{E}^{(p)\perp}$ and taking

$$\tilde{\mathbf{Q}} = (\tilde{\mathbf{U}}^{(p)})^+ \left[\tilde{\mathbf{U}}_2^{(p)\mathsf{T}}, \tilde{\mathbf{U}}_3^{(p)\mathsf{T}}, \cdots, \tilde{\mathbf{U}}_n^{(p)\mathsf{T}}, \mathbf{E}_{n+1}^{(p)\mathsf{T}}\right]^\mathsf{T}.$$

Given $\tilde{\mathbf{Q}}$, the new optimal values for $\mathbf{E}_i^{(p)}$ are obtained and the process iterated till convergence.

## 3 Pre-training of Recurrent Neural Networks

Here we define our pre-training procedure for recurrent neural networks with one hidden layer of $p$ units, and $O$ output units:

$$\mathbf{o}_t = \sigma(\mathbf{W}_{output} \mathbf{h}(\mathbf{x}_t)) \in \mathbb{R}^O, \quad \mathbf{h}(\mathbf{x}_t) = \sigma(\mathbf{W}_{input} \mathbf{x}_t + \mathbf{W}_{hidden} \mathbf{h}(\mathbf{x}_{t-1})) \in \mathbb{R}^p \tag{6}$$

where $\mathbf{W}_{output} \in \mathbb{R}^{O \times p}$, $\mathbf{W}_{hidden} \in \mathbb{R}^{p \times k}$, for a vector $\mathbf{z} \in \mathbb{R}^m$, $\sigma(\mathbf{z}) = [\sigma(z_1), \ldots, \sigma(z_m)]^\mathsf{T}$, and here we consider the symmetric sigmoid function $\sigma(z_i) = \frac{1-e^{-z_i}}{1+e^{-z_i}}$.

The idea is to exploit the hidden state representation obtained by eqs. (2) as initial hidden state representation for the RNN described by eqs. (6). This is implemented by initialising the weight matrices $\mathbf{W}_{input}$ and $\mathbf{W}_{hidden}$ of (6) by using the matrices that jointly solve eqs. (2) and eqs. (3), i.e. $\mathbf{A}$ and $\mathbf{B}$ (since $\mathbf{C}$ is function of $\mathbf{A}$ and $\mathbf{B}$). Specifically, we initialize $\mathbf{W}_{input}$ with $\mathbf{A}$, and $\mathbf{W}_{hidden}$ with $\mathbf{B}$. Moreover, the use of symmetrical sigmoidal functions, which do give a very good approximation of the identity function around the origin, allows a good transferring of the linear dynamics inside

RNN. For what concerns $\mathbf{W}_{output}$, we initialise it by using the best possible solution, i.e. the pseudoinverse of $\mathbf{H}$ times the target matrix $\mathbf{T}$, which does minimise the output squared error. Learning is then used to introduce nonlinear components that allow to improve the performance of the model. More formally, let consider a prediction task where for each sequence $\mathbf{s}_q \equiv (\mathbf{x}_1^q, \mathbf{x}_2^q, \ldots, \mathbf{x}_{l_q}^q)$ of length $l_q$ in the training set, a sequence $\mathbf{t}_q$ of target vectors is defined, i.e. a training sequence is given by $\langle \mathbf{s}_q, \mathbf{t}_q \rangle \equiv \langle (\mathbf{x}_1^q, \mathbf{t}_1^q), (\mathbf{x}_2^q, \mathbf{t}_2^q), \ldots, (\mathbf{x}_{l_q}^q, \mathbf{t}_{l_q}^q) \rangle$, where $\mathbf{t}_i^q \in \mathbb{R}^O$. Given a training set with $N$ sequences, let define the target matrix $\mathbf{T} \in \mathbb{R}^{L \times O}$, where $L = \sum_{q=1}^N l_q$, as $\mathbf{T} = \begin{bmatrix} \mathbf{t}_1^1, \mathbf{t}_2^1, \ldots, \mathbf{t}_{l_1}^1, \mathbf{t}_1^2, \ldots, \mathbf{t}_{l_N}^N \end{bmatrix}^\mathsf{T}$. The input matrix $\Xi$ will have size $L \times k$. Let $p^*$ be the desired number of hidden units for the recurrent neural network (RNN). Then the pre-training procedure can be defined as follows: *i)* compute the linear autoencoder for $\Xi$ using $p^*$ principal directions, obtaining the optimal matrices $\mathbf{A}^* \in \mathbb{R}^{p^* \times k}$ and $\mathbf{B}^* \in \mathbb{R}^{p^* \times p^*}$; *i)* set $\mathbf{W}_{input} = \mathbf{A}^*$ and $\mathbf{W}_{hidden} = \mathbf{B}^*$; *iii)* run the RNN over the training sequences, collecting the hidden activities vectors (computed using symmetrical sigmoidal functions) over time as rows of matrix $\mathbf{H} \in \mathbb{R}^{L \times p^*}$; *iv)* set $\mathbf{W}_{output} = \mathbf{H}^+ \mathbf{T}$, where $\mathbf{H}^+$ is the (left) pseudoinverse of $\mathbf{H}$.

## 3.1 Computing an approximate solution for large datasets

In real world scenarios the application of our approach may turn difficult because of the size of the data matrix. In fact, stable computation of principal directions is usually obtained by SVD decomposition of the data matrix $\Xi$, that in typical application domains involves a number of rows and columns which is easily of the order of hundreds of thousands. Unfortunately, the computational complexity of SVD decomposition is basically cubic in the smallest of the matrix dimensions. Memory consumption is also an important issue. Algorithms for approximate computation of SVD have been suggested (e.g., [18]), however, since for our purposes we just need matrices $\mathbf{V}$ and $\mathbf{\Lambda}$ with a predefined number of columns (i.e. $p$), here we present an ad-hoc algorithm for approximate computation of these matrices. Our solution is based on the following four main ideas: *i)* divide $\Xi$ in slices of $k$ (i.e., size of input at time $t$) columns, so to exploit SVD decomposition at each slice separately; *ii)* compute approximate $\mathbf{V}$ and $\mathbf{\Lambda}$ matrices, with $p$ columns, incrementally via truncated SVD of temporary matrices obtained by concatenating the current approximation of $\mathbf{V\Lambda}$ with a new slice; *iii)* compute the SVD decomposition of a temporary matrix via either its kernel or covariance matrix, depending on the smallest between the number of rows and the number of columns of the temporary matrix; *iv)* exploit QR decomposition to compute SVD decomposition.

Algorithm 1 shows in pseudo-code the main steps of our procedure. It maintains a temporary matrix $\mathbf{T}$ which is used to collect incrementally an approximation of the principal subspace of dimension $p$ of $\Xi$. Initially (line 4) $\mathbf{T}$ is set equal to the last slices of $\Xi$, in a number sufficient to get a number of columns larger than $p$ (line 2). Matrices $\mathbf{V}$ and $\mathbf{\Lambda}$ from the $p$-truncated SVD decomposition of $\mathbf{T}$ are computed (line 5) via the KECO procedure, described in Algorithm 2, and used to define a new $\mathbf{T}$ matrix by concatenation with the last unused slice of $\Xi$. When all slices are processed, the current $\mathbf{V}$ and $\mathbf{\Lambda}$ matrices are returned. The KECO procedure, described in Algorithm 2 , reduces the computational burden by computing the $p$-truncated SVD decomposition of the input matrix $\mathbf{M}$ via its kernel matrix (lines 3-4) if the number of rows of $\mathbf{M}$ is no larger than the number of columns, otherwise the covariance matrix is used (lines 6-8). In both cases, the $p$-truncated SVD decomposition is implemented via QR decomposition by the INDIRECTSVD procedure described in Algorithm 3. This allows to reduce computation time when large matrices must be processed [19]. Finally, matrices $\mathbf{V}$ and $\mathbf{S}^{\frac{1}{2}}$ (both kernel and covariance matrices have squared singular values of $\mathbf{M}$) are returned.

We use the strategy to process slices of $\Xi$ in reverse order since, moving versus columns with larger indices, the rank as well as the norm of slices become smaller and smaller, thus giving less and less contribution to the principal subspace of dimension $p$. This should reduce the approximation error cumulated by dropping the components from $p + 1$ to $p + k$ during computation [20]. As a final remark, we stress that since we compute an approximate solution for the principal directions of $\Xi$, it makes no much sense to solve the problem given in eq. (5): learning will quickly compensate for the approximations and/or sub-optimality of $\mathbf{A}$ and $\mathbf{B}$ obtained by matrices $\mathbf{V}$ and $\mathbf{\Lambda}$ returned by Algorithm 1. Thus, these are the matrices we have used for the experiments described in next section.

**Algorithm 1** Approximated $\mathbf{V}$ and $\mathbf{\Lambda}$ with $p$ components

1: **function** SVFORBIGDATA($\mathbf{\Xi}, k, p$)
2:     $nStart = \lceil p/k \rceil$                                                            ▷ Number of starting slices
3:     $nSlice = (\mathbf{\Xi}.columns/k) - nStart$                               ▷ Number of remaining slices
4:     $\mathbf{T} = \mathbf{\Xi}[:, k * nSlice : \mathbf{\Xi}.columns]$
5:     $\mathbf{V}, \mathbf{\Lambda} =$ KECO($\mathbf{T}, p$)                          ▷ Computation of $\mathbf{V}$ and $\mathbf{\Lambda}$ for starting slices
6:     **for** $i$ in REVERSED(range($nSlice$)) **do**      ▷ Computation of $\mathbf{V}$ and $\mathbf{\Lambda}$ for remaining slices
7:         $\mathbf{T} = [\mathbf{\Xi}[:, i * k:(i + 1) * k], \mathbf{V}\mathbf{\Lambda}]$
8:         $\mathbf{V}, \mathbf{\Lambda} =$ KECO($\mathbf{T}, p$)
9:     **end for**
10:     **return** $\mathbf{V}, \mathbf{\Lambda}$
11: **end function**

---

**Algorithm 2** Kernel vs covariance computation

1: **function** KECO($\mathbf{M}, p$)
2:     **if** $\mathbf{M}.rows <= \mathbf{\Xi}.columns$ **then**
3:         $\mathbf{K} = \mathbf{M}\mathbf{M}^{\mathsf{T}}$
4:         $\mathbf{V}, \mathbf{S}_{sqr}, \mathbf{U}^{\mathsf{T}} =$ INDIRECTSVD($\mathbf{K}, p$)
5:     **else**
6:         $\mathbf{C} = \mathbf{M}^{\mathsf{T}}\mathbf{M}$
7:         $\mathbf{V}, \mathbf{S}_{sqr}, \mathbf{U}^{\mathsf{T}} =$ INDIRECTSVD($\mathbf{C}, p$)
8:         $\mathbf{V} = \mathbf{M}\mathbf{U}^{\mathsf{T}}\mathbf{S}_{sqr}^{-\frac{1}{2}}$
9:     **end if**
10:     **return** $\mathbf{V}, \mathbf{S}_{sqr}^{\frac{1}{2}}$
11: **end function**

**Algorithm 3** Truncated SVD by QR

1: **function** INDIRECTSVD($\mathbf{M}, p$)
2:     $\mathbf{Q}, \mathbf{R} =$ QR($\mathbf{M}$)
3:     $\mathbf{V}_r, \mathbf{S}, \mathbf{U}^{\mathsf{T}} =$ SVD($\mathbf{R}$)
4:     $\mathbf{V} = \mathbf{Q}\mathbf{V}_r$
5:     $\mathbf{S} = \mathbf{S}[1 : p, 1 : p]$
6:     $\mathbf{V} = \mathbf{V}[1 : p, :]$
7:     $\mathbf{U}^{\mathsf{T}} = \mathbf{U}^{\mathsf{T}}[:, 1 : p]$
8:     **return** $\mathbf{V}, \mathbf{S}, \mathbf{U}^{\mathsf{T}}$
9: **end function**

## 4 Experiments

In order to evaluate our pre-training approach, we decided to use the four polyphonic music sequences datasets used in [21] for assessing the prediction abilities of the RNN-RBM model. The prediction task consists in predicting the notes played at time $t$ given the sequence of notes played till time $t - 1$. The RNN-RBM model achieves state-of-the-art in such demanding prediction task. As performance measure we adopted the accuracy measure used in [21] and described in [22]. Each dataset is split in training set, validation set, and test set. Statistics on the datasets, including largest sequence length, are given in columns 2-4 of Table 1. Each sequence in the dataset represents a song having a maximum polyphony of 15 notes (average 3.9); each time step input spans the whole range of piano from A0 to C8 and it is represented by using 88 binary values (i.e. $k = 88$).

Our pre-training approach (PreT-RNN) has been assessed by using a different number of hidden units (i.e., $p$ is set in turn to 50, 100, 150, 200, 250) and 5000 epochs of RNN training[1] using the Theano-based stochastic gradient descent software available at [23].

Random initialisation (Rnd) has also been used for networks with the same number of hidden units. Specifically, for networks with 50 hidden units, we have evaluated the performance of 6 different random initialisations. Finally, in order to verify that the nonlinearity introduced by the RNN is actually useful to solve the prediction task, we have also evaluated the performance of a network with linear units (250 hidden units) initialised with our pre-training procedure (PreT-Lin250).

To give an idea of the time performance of pre-training with respect to the training of a RNN, in column 5 of Table 1 we have reported the time in seconds needed to compute pre-training matrices (Pre-) (on Intel© Xeon© CPU E5-2670 @2.60GHz with 128 GB) and to perform training of a RNN with $p = 50$ for 5000 epochs (on GPU NVidia K20). Please, note that for larger values of $p$, the increase in computation time of pre-training is smaller than the increment in computation time needed for training a RNN.

| Dataset | Set | # Samples | Max length | (Pre-)Training Time | Model | ACC% [21] |
|---|---|---|---|---|---|---|
| Nottingham | Training ($39165 \times 56408$) | 195 | 641 | seconds (226) 5837 | RNN (w. HF) | 62.93 (66.64) |
| | | | | | RNN-RBM | **75.40** |
| | Test | 170 | 1495 | $p = 50$ | PreT-RNN | 75.23 ($p = 250$) |
| | Validation | 173 | 1229 | 5000 epochs | PreT-Lin250 | 73.19 |
| Piano-midi.de | Training ($70672 \times 387640$) | 87 | 4405 | seconds (2971) 4147 | RNN (w. HF) | 19.33 (23.34) |
| | | | | | RNN-RBM | 28.92 |
| | Test | 25 | 2305 | $p = 50$ | PreT-RNN | **37.74** ($p = 250$) |
| | Validation | 12 | 1740 | 5000 epochs | PreT-Lin250 | 16.87 |
| MuseData | Training ($248479 \times 214192$) | 524 | 2434 | seconds (7338) 4190 | RNN (w. HF) | 23.25 (30.49) |
| | | | | | RNN-RBM | 34.02 |
| | Test | 25 | 2305 | $p = 50$ | PreT-RNN | **57.57** ($p = 200$) |
| | Validation | 135 | 2523 | 5000 epochs | PreT-Lin250 | 3.56 |
| JSB Chorales | Training ($27674 \times 22792$) | 229 | 259 | seconds (79) 6411 | RNN (w. HF) | 28.46 (29.41) |
| | | | | | RNN-RBM | 33.12 |
| | Test | 77 | 320 | $p = 50$ | PreT-RNN | **65.67** ($p = 250$) |
| | Validation | 76 | 289 | 5000 epochs | PreT-Lin250 | 38.32 |

Table 1: Datasets statistics including data matrix size for the training set (columns 2-4), computational times in seconds to perform pre-training and training for 5000 epochs with $p = 50$ (column 5), and accuracy results for state-of-the-art models [21] vs our pre-training approach (columns 6-7). The acronym (w. HF) is used to identify an RNN trained by Hessian Free Optimization [4].

Training and test curves for all the models described above are reported in Figure 1. It is evident that random initialisation does not allow the RNN to improve its performance in a reasonable amount of epochs. Specifically, for random initialisation with $p = 50$ (Rnd 50), we have reported the average and range of variation over the 6 different trails: different initial points do not change substantially the performance of RNN. Increasing the number of hidden units allows the RNN to slightly increase its performance. Using pre-training, on the other hand, allows the RNN to start training from a quite favourable point, as demonstrated by an early sharp improvement of performances. Moreover, the more hidden units are used, the more the improvement in performance is obtained, till overfitting is observed. In particular, early overfitting occurs for the Muse dataset. It can be noticed that the linear model (Linear) reaches performances which are in some cases better than RNN without pre-training. However, it is important to notice that while it achieves good results on the training set (e.g. JSB and Piano-midi), the corresponding performance on the test set is poor, showing a clear evidence of overfitting. Finally, in column 7 of Table 1, we have reported the accuracy obtained after validation on the number of hidden units and number of epochs for our approaches (PreT-RNN and PreT-Lin250) versus the results reported in [21] for RNN (also using Hessian Free Optimization) and RNN-RBM. In any case, the use of pre-training largely improves the performances over standard RNN (with or without Hessian Free Optimization). Moreover, with the exception of the Nottingham dataset, the proposed approach outperforms the state-of-the-art results achieved by RNN-RBM. Large improvements are observed for the Muse and JSB datasets. Performance for the Nottingham dataset is basically equivalent to the one obtained by RNN-RBM. For this dataset, also the linear model with pre-training achieves quite good results, which seems to suggest that the prediction task for this dataset is much easier than for the other datasets. The linear model outperforms RNN without pre-training on Nottingham and JSB datasets, but shows problems with the Muse dataset.

# 5 Conclusions

We have proposed a pre-training technique for RNN based on linear autoencoders for sequences. For this kind of autoencoders it is possible to give a closed form solution for the definition of the "optimal" weights, which however, entails the computation of the SVD decomposition of the full data matrix. For large data matrices exact SVD decomposition cannot be achieved, so we proposed a computationally efficient procedure to get an approximation that turned to be effective for our goals. Experimental results for a prediction task on datasets of sequences of polyphonic music show the usefulness of the proposed pre-training approach, since it allows to largely improve the state of the art results on all the considered datasets by using simple stochastic gradient descend for learning. Even if the results are very encouraging the method needs to be assessed on data from other application domains. Moreover, it is interesting to understand whether the analysis performed in [24] on linear deep networks for vectors can be extended to recurrent architectures for sequences and, in particular, to our method.

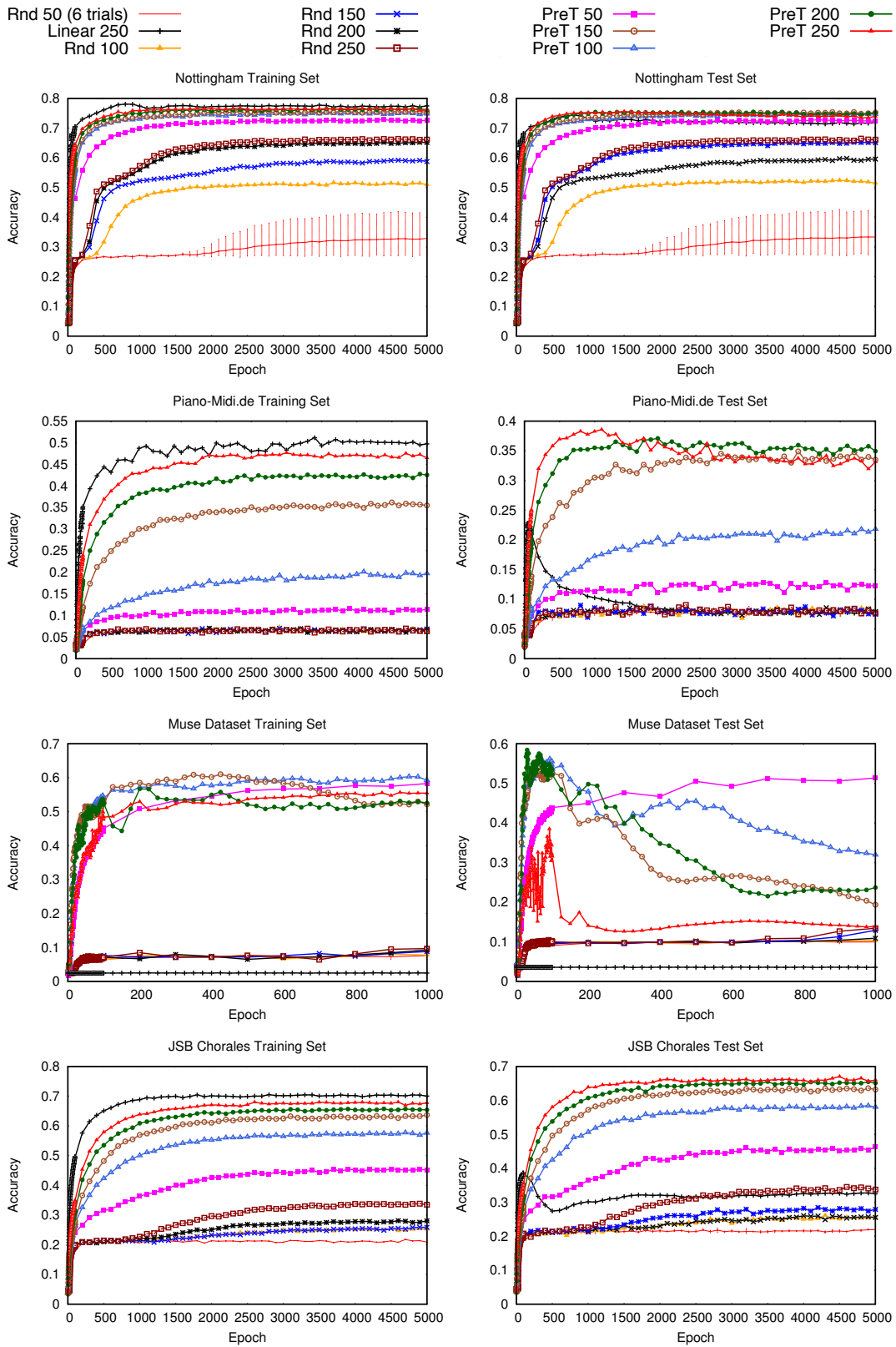

Figure 1: Training (left column) and test (right column) curves for the assessed approaches on the four datasets. Curves are sampled at each epoch till epoch 100, and at steps of 100 epochs afterwards.

## Footnotes

[1]Due to early overfitting, for the Muse dataset we used 1000 epochs.

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
