[Reviews · NeurIPS 2014]

Submitted by Assigned_Reviewer_9

Quality and originality: The paper is well written. The introduction motivates the general problem. Section 2 is very easy to read, providing both explanation and necessary linear algebra. and does a good job of supporting the proposed model. I see two important issues in the paper. First is the motivation of the importance of pre-training. The authors are convincing here. The second is whether they authors’ particular version of pre-training for RNNs is the right one. Rather they show that they can outperform a particular RNN approach which uses no pre-training. Here it’s a bit more confusing, as they compare only to the results from a single paper on music data. Furthermore there’s some confusion as to how much better the proposed model works compared to the one from Sperduti in 2007 (the “conceptual framework” extended here by the authors). Does the new model actually work better than the Sperduti one on music data? Or is it just that the theoretical approach is reworked? The authors address this directly in the closing paragraph of Section 1, but I wish they had spent more time pulling apart their contributions from those of previous work on pre-training. Furthermore, I wish they had at least implemented Sperduti’s method in the experiments section.

Clarity and Significance: It’s not clear why the authors chose to use music datasets. No analyses of the performance of the musical performance of the models is given, nor do we get any musical examples. This adds some confusion to the paper: is this about sequence prediction in general or about music sequence prediction? I presume it’s about sequence prediction in general, but would have appreciated either some non-music benchmarks or some more exploration of the music space. Of course space is an issue. However I would have preferred that the lemmas and some of the algorithmic details be moved out to supporting material, with more time spent exploring how pre-training does anything other than raise accuracy.
Summary: This paper looks at the effectiveness of pre training RNNs using linear auto encoders for sequences. The method, and its relationship to PCA, is provided, as well as an approximate method suitable to large datasets based on SVD decomposition. Results for four music-related sequence learning tasks are presented and discussed.

Submitted by Assigned_Reviewer_14

1) Not clear the relation between eq. (2) and (3) and (6)

2) Why the authors assume that \Sigma=U{p}?

3) In the literature of linear systems (e.g., Kalman filter) there are many closed for solutions to find the parameters of the linear model, for instance
State-Space System Identification with Identified Hankel Matrix (http://www.dartmouth.edu/~mqphan/Resources/TP3045.pdf)

4) The authors pre-train the RNN assuming it is a linear dynamic autoencoder, but they apply a non-linearity later. What is the error with the original pre-training (without improving the error later)?

Over an interesting paper, but some comparisons with standard pre-training methods should be done in deep learning. For instance, training independently Winput, Whidden and Woutput.

Summary: The paper proposes a closed form solution to train a linear recurrent network, and use this parameter as a pre-training for a non-linear RNN.

Submitted by Assigned_Reviewer_26

The paper proposes a method for pre-training recurrent neural networks with the solution of a linear dynamic system. Experiments show that even an approximate solution to this system yields impressive results on a music prediction task.

Quality
The paper is technically sound and empirical results bolster claims.

Clarity
The paper is organized and written well. Algorithms are sufficiently motivated and documented.

Originality
This appears to mesh with the ICLR 2014 paper "Exact solutions to the nonlinear dynamics of learning in deep linear neural networks" by Saxe et al. . It should be cited and discussed.

Significance
The theoretical and empirical results are significant. The algorithms are readily usable by practitioners and extensible by other researchers.
Summary: The paper is lucid and presents a fascinating and significant result that could change the way researchers train recurrent neural networks.
Author Feedback
Author rebuttal: Reviewer 1:
1) Equations (2) and (3) define the dynamics of a linear autoencoder network applied to sequential data. The idea is to exploit the hidden state representation obtained by eq. (2) as initial hidden state representation for the RNN described by eq. (6). This is implemented by initialising the weight matrices W_input and W_hidden of eq. (6) by using the matrices that jointly solve equations (2) and (3), i.e. A and B (since C is function of A and B). Specifically, we initialize W_input with A, and W_hidden with B (see step 2 of pre-training procedure in section 3). Moreover, the use of symmetrical sigmoidal functions, which do give a very good approximation of the identity function around the origin, allows a good transfering of the linear dynamics inside RNN.
For what concerns W_output, we initialise it by using the best possible solution, i.e.
the pseudoinverse of H times the target matrix T, which does minimise the output squared error.
Learning is then used to introduce nonlinear components that allow to improve the performance of the model.

2) The reason to impose Omega = U^{(p)} is to get a state space where the coordinates are uncorrelated so to diagonalise the empirical sample covariance matrix of the states, i.e. Y^T Y. Please, note that in this way each state (i.e., row of the Y matrix) corresponds to a row of the data matrix \Xi, i.e. the unrolled (sub)sequence read up to a given time t. If the rows of \Xi were vectors, this would correspond to compute PCA, allowing to minimize the squared reconstruction error obtained by an autoencoder output, with p < rank(\Xi) hidden units, by dropping the less important coordinates, i.e. the ones corresponding to the smallest singular values. However, in Section 2.1, we show that this is not the case for our setting where, given a specific state space dimension p < rank(\Xi), the optimal reconstruction value is obtained by solving the problem described in eq. (5). In fact, when p < rank(\Xi), since the reconstructed \Xi is different from the actual \Xi, the equation defining matrix Q, introduced at page 3, line 133, does not hold anymore.

3) It is true that other approaches can be used and should be further explored,
however in standard system identification the state space dynamics is devised in function of the desired output, making it difficult to use the linear model for pre-training: injecting the weights of the linear system into an RNN does not necessarily preserve the linear solution due to the nonlinearity of RNN; moreover, successive training of RNN does tend to undo the work done by the adopted system identification technique. In fact, no pre-training procedure for RNN based on this ground has been proposed in the past. In our formulation, the state space representation is only dependent on the input data (i.e., it is unsupervised), allowing the exploitation of dynamical systems of different sizes to initialise the RNN at different levels of approximation. Basically, the trick is to propagate forward in time as much information as possibile of the input sequence.
By the way, the reference cited by the reviewer is based as well on SVD decomposition...

4) The accuracy after the pre-training phase (and before training) is represented by the first point (epoch 0) of each learning curve in the plots at page 8. Indeed each plot shows the evolution of accuracy from epoch 0 of training until its end.

Concerning the comparison versus other pre-training approaches, while it is true that there exist approaches for DEEP networks involving VECTORS in input, no pre-training procedure has been proposed in literature for RNN processing SEQUENCES. A somewhat related pre-training procedure for sequences has been specifically defined and used into the RNN-RBM model [20], however this procedure is
not applicable to RNNs, and the accuracy returned by RNN-RBM is significantly inferior with respect to the one obtained by our proposal in 3 out of 4 used datasets.

Reviewer 2
The referenced ICLR 2014 paper is very recent and surely needs to be cited and discussed in the paper!

Reviewer 3
The previous work by Sperduti is limited to the solution of equations (2) and (3). In the cited papers [13,14,15] there is no attempt to apply the theory to perform pre-training of a RNN. So there is no pre-training technique by Sperduti versus which to compare our approach.
By stretching it a lot, the linear model discussed in the paper could be considered the closest one to the work by Sperduti. However, the linear model contains an output layer for prediction that is: i) completely different from the one described in eq. (3); (ii) never considered in the work by Sperduti.
Concerning the use of musical datasets, indeed our aim was to develop a general approach for pre-training RNN.
We decided to test it on music sequences because the prediction task for this type of data is very challenging, for several reasons. In fact, dealing with music data is hard, due to the input size at each time step and to the length of each single sequence. These issues make it difficult to train a RNN, and thus necessary to adopt a pre-training approach.